

# Role of air-soil temperature on the LAI course and role of height-DBH on the maximum LAI during foliation of *Platanus orientalis* L. along an urban-rural greenway system

Melih Öztürk[1], Turgay Biricik[1], Rıdvan Koruyan[1]

[1]Department of Landscape Architecture, Faculty of Engineering, Architecture and Design, Bartın University, Bartın, 74110, Turkey

*Correspondence to*: Melih Öztürk (melihozturk@bartin.edu.tr)

**Abstract.** Rural greenway systems passing through woodlands to connect urban societies, are valuable in terms of not only transportation but also roadside tree phenology and ecophysiology, and associated recreation. Therefore, particularly during their foliation periods, monitoring and analyses of that phenological and eco-physiological course of these roadside trees referring to the significant and comprehensive canopy parameter; LAI, primarily will indicate their gradual degree of closure, and will determine their gradual coverage on the road and the roadside. This gradual closure indicator and coverage determinant parameter can further be used for detecting shading and recreation potential, and safety level of those greenways. Nevertheless, major driving factors on that phenological and eco-physiological course, can also be investigated by monitoring and assessing the development and change of the mean LAI under the influence of the mean temperature, height and DBH values. Therefore, for this study, in order to monitor and determine the development and change of the LAI, hemispherical photographs were taken beneath the tree canopies at 10 different points along the part of a well-known greenway system, which involves alleys of *Platanus orientalis* L. (oriental plane) trees. This point-based hemispherical photographing procedure was applied and repeated 20 times particularly during the foliation period between the mid-March and late-June, when totally 200 photographs were obtained and analysed using digital image processing method. The seasonal course of the LAI values was graphed for each point and principally the daily-based mean LAI (ranging between 0.35 and 2.76 $m^2$ $m^{-2}$) was evaluated referring to both the air and soil (-10 cm) temperature data. The point-based maximum LAI values (average 2.76 $m^2$ $m^{-2}$; ranging between 2.42 $m^2$ $m^{-2}$ and 3.16 $m^2$ $m^{-2}$), which were achieved during the mid-June, were examined comparing their ranking order with those of the basic physiological parameters; mean height (ranging between 17.0 m and 26.7 m) and mean DBH (ranging between 26.5 cm and 38.2 cm), and number of trees (5 to 15) within the canopy frames of the relevant points. Afterwards, the phenological-based and daily-based mean LAI values were discussed dependent upon their high and significant correlation particularly with the soil temperature data ($r$=0.89, $P$<0.01), whereas the point-based maximum LAI values were also discussed dependent upon their non-correlation with the point-based mean height and mean DBH. In conclusion, the overall results of this study primarily emphasized the influence of the soil temperature on the phenological course of the oriental plane canopies and on the development of their daily-based mean LAI particularly during their foliation period. This current effect of the soil temperature indicated the potential alarm of the early budburst dates and associated possible advance of the tree foliation





period, dependent on the warming capacity of the road asphalt and roadside pavement on the soil underneath, particularly during and after the newly pavement and resurfacing practices.

**Keywords:** Greenway system; roadside trees; oriental plane; air-soil temperature, height; DBH; LAI

## 1 Introduction

Introduction of intercity roads into the rural landscapes, which are composed particularly of the vegetative ecosystems, already impose ecological sensitivity, and therefore incur environmental responsibility (Forman and Alexander, 1998). These intercity roads that pass throughout the woodlands and forests mostly constitute rural greenway systems, ultimately become part of the overall rural landscape by integrating with those surrounding wooded ecosystems (Konijnendijk, 2008; Turner and Gardner, 2015). Their trees frequently lined up on the roadside often form alleys with different canopy closures throughout the different seasons, and hence play active roles in road shading and safety, and associated recreation (Wolf, 2003). Species, arrangement and number of these trees together with their physiological characteristics are the common and key factors for determining the degree of canopy closure, and relevant coverage over their projection on the road and roadside ground (Rahman et al., 2015). However, principally for the deciduous tree canopies, phenological course of these roadside trees, particularly during their foliation periods, also influence to differentiate the degree of their canopy closure, and relevant coverage not only on a spatial basis but also on a temporal basis (Neyns and Canters, 2022). Therefore, continuous monitoring on the purpose of determining both their phenological course and physiological characteristics through the analyses of certain vegetation parameters, will as well help to define the degree of their canopy closure and relevant coverage (Granero-Belinchon, 2020; Pu, 2021) on the road and roadsides of those rural greenway systems (Wang et al., 2019). Furthermore, associating and correlating those certain parameters with some of the ecological factors such as daily-based temperature data will also contribute to discover their eco-physiological characteristics, which will consequently assist their probable role for ecosystem services (García-Pardo et al., 2022).

Dependent principally upon their active role on the photosynthesis, leaves and canopies they form, are effective on stem physiology of the trees (Pretzsch, 2009). Thus, diameter at breast height (DBH) and height of the trees are the two significant physiological parameters that define their growth characteristics (Landsberg and Sands, 2011). Hence, stem and overall tree physiology is directly related with that canopy physiology and characteristics. However, the reverse role of the tree leaves and canopies on their DBH and height, has not been thoroughly determined particularly for the deciduous trees. Although there are scientific studies that have tried to correlate crown area (e.g., Blanchard et al., 2016) and canopy structure (Côté et al., 2012) of the trees with their height and DBH, tree canopy parameters that are directly referred as indicators of the canopy characteristics under the influence of those stem physiological characteristics, are relatively restricted. Out of those canopy parameters, Leaf Area Index (LAI; $m^2\ m^{-2}$), that defines one-sided area of all the leaves within a particular canopy over the projection area of that canopy (Burrows et al., 2002), is the most prominent one. Therefore, LAI is the key parameter indicating the influence level of the leaves and canopies of the tree-shrub communities, within many substantial phenological, ecological,



eco-physiological, climatological, and hydrological processes (Bonan, 2016). Nonetheless, the LAI parameter is accordingly
an indicator of canopy closure, and associated shading potential of those tree canopies, ultimately the recreation potential of
that environment where those trees exist (Zhang et al., 2022).  Moreover, many recent studies have indicated the relatively
high and significant positive correlations between the LAI and the tree and canopy heights (Yuan et al., 2013; Klingberg et al.,
2017). On the other hand, air and soil temperatures are the main driving climatic factors on the phenological and eco-
physiological processes, and hence to some extent on the periodical, seasonal, and overall annual course of the LAI values of
the forest trees within both the urban and rural landscapes (Öztürk et al., 2015).

Consequently, the main aim of this study is to investigate and reveal the priority and effective role of the air-soil temperature
on the seasonal change of the LAI trees, and to question the existence of any role of the tree physiology on the maximum LAI
values along a roadside. Thus, in this study, LAI course of the oriental plane (*Platanus orientalis* L.) trees was monitored and
analysed principally throughout the foliation seasons particularly referring to the daily-based relevant temperature data and
referring to their physiological parameters during solely the maximum LAI date. For this purpose, LAIs of the tree canopies
over the definite different points were determined along the part of a greenway system, which constitutes a certain section of
a temperate forest ecosystem between two provinces. These determined LAI values, and their correlation with the air-soil
temperature data, were discussed with regarding to their phenological periods. Besides, the point-based maximum LAI values
were discussed based primarily on their height and DBH values, especially questioning the existence and level of correlation
with them. By this way, the influence of the tree stem physiology on the maximum LAI values of the tree canopies would have
been examined.

## 1.1 Species description of *Platanus orientalis* L.

*Platanus orientalis* L. (oriental plane), which belong to the Platanaceae family, is a deciduous vigorous tree up to 30 m height,
overshadowing large area with its' large canopy (Boyd et al., 1996). Their massive trunk is smooth with ash-grey bark, which
peels off in large thin flakes whereas their large leaves with flat surface are cut into five deep lobes, with numerous secondary
notches (Johns, 2014). They are both forest and ornamental trees, which naturally extend from the Balkans towards the
Anatolia, and then towards the Himalayas (Davis, 1982). However, they also exist in Ireland and South America (Johns, 2014).
They require direct sunlight and sunny environment, and therefore need warm and humid climates (Çepel, 1994) whereas they
prefer good soil on high water table, and near running water at the valley bottoms (Saatçioğlu, 1976). Given the fact that their



physiological characteristics particularly their height and vast canopy led them to be among the distinguished alley trees, this
also made them suitable for the urban and rural greenways (Johns, 2014).

**Figure 1: Location of the focal study field together with the hemispherical photographing points on the intercity road and overall greenway system within Turkey. Basemaps are derived from ArcMap10.5 (ESRI Inc., 2016).**





## 2 Material and Methodology

### 2.1 Study Site Characteristics and Hemispherical Photographing Points

The study site is part of the greenway system between the Bartın and Karabük Provinces of Western Black Sea Region of Turkey (Fig. 1). The study site is approximately 15 km away from the Abdipaşa Town and about 40 km away from its' Bartın Province, both at the northwest, where there are frequent district and village settlements along the road (Fig. 1). On the other side, the study site is approximately 40 km away from the World Heritage City, Safranbolu (UNESCO, 1994), and about 50 km away from its' Karabük Province, both at the south, where there are also frequent district and village settlements along the road (Fig. 1). Therefore, the greenway system along this intercity road, which is famous primarily from the point of spectacular tree canopy tunnels, quite dense in some places. In order to secure the sustainability of this greenway system, road construction and repair works are carried out very sensitively, and hence monitoring studies are encouraged.

Thus, the study site consists of 10 hemispherical photographing points on the centre of the single-lane road beneath the canopies of the *Platanus orientalis* L. (oriental plane) trees (Fig. 1). Explicitly, the study site is located between the 41° 27' 22" and 41° 27' 35" northern latitudes and, between the 32° 41' 1" and 32° 41' 17" eastern longitudes (Fig. 1). The extent of part of the road, which was chosen as the focal research area is approximately 700 m (Fig. 1), where the average altitude is 290 m asl. The intervals between the hemispherical photographing points change between 20 m and 300 m, which were the ultimate choice reasons for best representing the differentiation of canopy characteristics within the focal research area (Fig. 1). Forest stands, agricultural fields, and fluvial sands constitute the dominant land uses surrounding the hemispherical photographing points on the road within the focal research area (Fig. 1). The forest stands, which are particularly scattered on slopes and tops of the surrounding hills, are mainly composed of black pines (*Pinus nigra* Arnold), European hornbeams (*Carpinus betulus* L.) oriental beeches (*Fagus orientalis* Lipsky), and sessile oaks (*Quercus petraea* [Matt.] Liebl.) (TGDF, 2021) (Figs. 1 and 2). On the other hand, some agricultural fields, which are principally at the foot of the hills and at the lowlands close to the stream, include few village houses. The altitude of those surrounding hills range between 390 m asl. and 490 m asl. The shallow (20-50 cm) or moderate deep (50-90 cm) grey-brown podzolic soils together with brown forest soils have formed particularly on the slope and top of the hills whereas deep (90+ cm) alluvial soils have been deposited at the lowlands close to the stream (TMAF, 2005). However, these soils have been composed and formed on sandstones and mudstones (TGDMRE, 2007). The research region drops within the mesothermal climate regime (Atalay, 2011) with approximately 1180 mm average annual total precipitation and 12.5°C average annual temperature (TSMS, 2022).



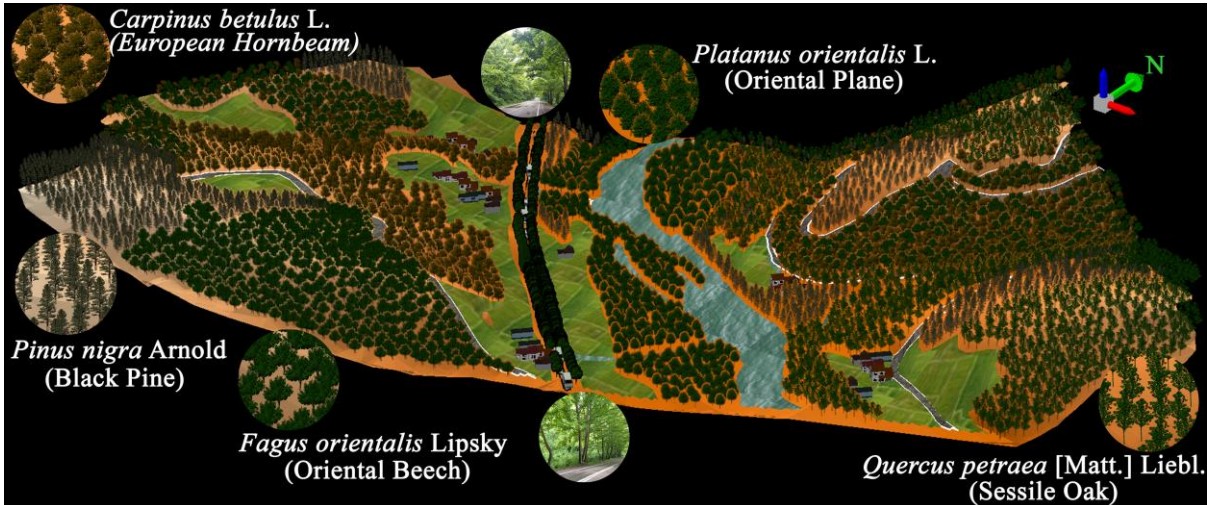

**Figure 2: Exact scaled 3D simulation and illustration of part of the overall greenway system with the surrounding forest and agricultural landscape focusing primarily on the study field together with the representative tree species (with their focal images), stream channel, village houses and roads, intercity road and cars**

## 2.2 Meteorological Calculations

The data of the six different meteorological stations, which were at the 36 m asl. (~33 km away at northwest; Bartın), at the 134 m asl. (~12 km away at northwest; Ulus), at the 359 m asl. (~5 km away at east; Ovacuma), at 520 m asl. (~12 km away at west; Ceyüpler), at the 540 m asl. (~29 km away at northeast; Çubukeli), at the 1090 m asl. (11 km away at south; Orman), were referred during the analyses and determination of the air temperature values for the study field as suggested by the scientific literature for such as this mesothermal humid region (TSMS, 2022). As a matter of fact, initially, correlation analyses were conducted between the air temperature data of these meteorological stations. Some of their correlation ($r$=0.96 correlation; $P$<0.01) and regression suggested the lapse rate of 0.5°C/100 m asl., as also suggested by Barry (2008) for the mesothermal humid regions. Since its' air temperature data were more consistent and soil temperature data were also measured there, the temperature data of Bartın Metetorological Station were used during the calculation of the temperature data for the study field (TSMS, 2022). Therefore, the air and soil (-10 cm) temperature data of this meteorological station were modified based on the suggested lapse rate and the altitudinal difference between the meteorological station and the study field. Dependent upon the densely distribution of the oriental plane tree roots within the 0 and 15 cm (Xie et al., 2020), the soil temperature data belonging to the -10 cm soil depth were referred. Provided that the 1-week air-soil (-10 cm) temperature data prior to the first monitoring in early-March were included, the modified mean air-soil temperature data throughout those frequent intervals between each field visits for hemispherical photographing, were calculated.



### 2.3 Hemispherical Photographing and LAI, Height, DBH Analyses

The 10 hemispherical photographing points on the middle lane of the road were chosen in the field and marked (Fig. 1). The hemispherical photographing (180°) of the tree canopies were conducted on these fixed points beneath and towards those canopies above. For this hemispherical photographing perpendicular (90°) to the middle lane of the road, an 8 mm fisheye objective (Sigma F3.5 EX DG Circular Fisheye-Sigma Corporation) was mounted on a digital camera (Canon EOS 5D SLR digital camera-Canon Inc.). Beginning from the 75[th] day (mid-March) of the year until the 209[th] day (late-July) of the year, 20 field visits were done, by which 200 hemispherical photographs were obtained in total (Fig. 3). Hence, in order to analyse the LAI values, the digital images of these 200 hemispherical photographs were processed by the software program; Hemisfer version 3.3 (Swiss Federal Institute of Forest, Snow and Landscape Research; Schleppi et al., 2007). The methodology by Thimonier et al. (2010) was principally referred during the analyses of LAI since it is more compatible for the deciduous trees. On the other hand, for the thresholding, automatic detection methodology suggested by the study of Nobis and Hunziker (2005), was applied. However, the necessary corrections were made referring to the integrated methodology dependent upon the studies by Schleppi et al. (2007), and Chen and Cihlar (1995). The number of oriental plane trees that are included within the canopy frame of each hemispherical photographing point was recorded. Thereby, the height and DBH of each of these plane trees within those frames, were measured and recorded according to their points, when they were foliated. Indeed, the heights of the oriental planes were measured using the Blume-Leiss ALTIMeter Model BL6 (Carl Leiss Slope and Tree Height Measuring Device) whereas their DBHs were measured using the tree calliper. The statistical correlation analyses between the interim mean air-soil temperature data and the mean LAI values were conducted referring to the Pearson correlation and significance tests (Devore and Farnum 1999) via SPSS software; the version 22.0 (SPSS Inc., Chicago, IL). The same correlation and significance tests were also applied for the statistical correlation analyses between the point-based mean tree height-DBH values and the point-based maximum LAI values.





**Figure 3: LAI values (y-axis) of all 10 points along the monitoring DOYs; 2023 (x-axis) together with their corresponding hemispherical photographs.**





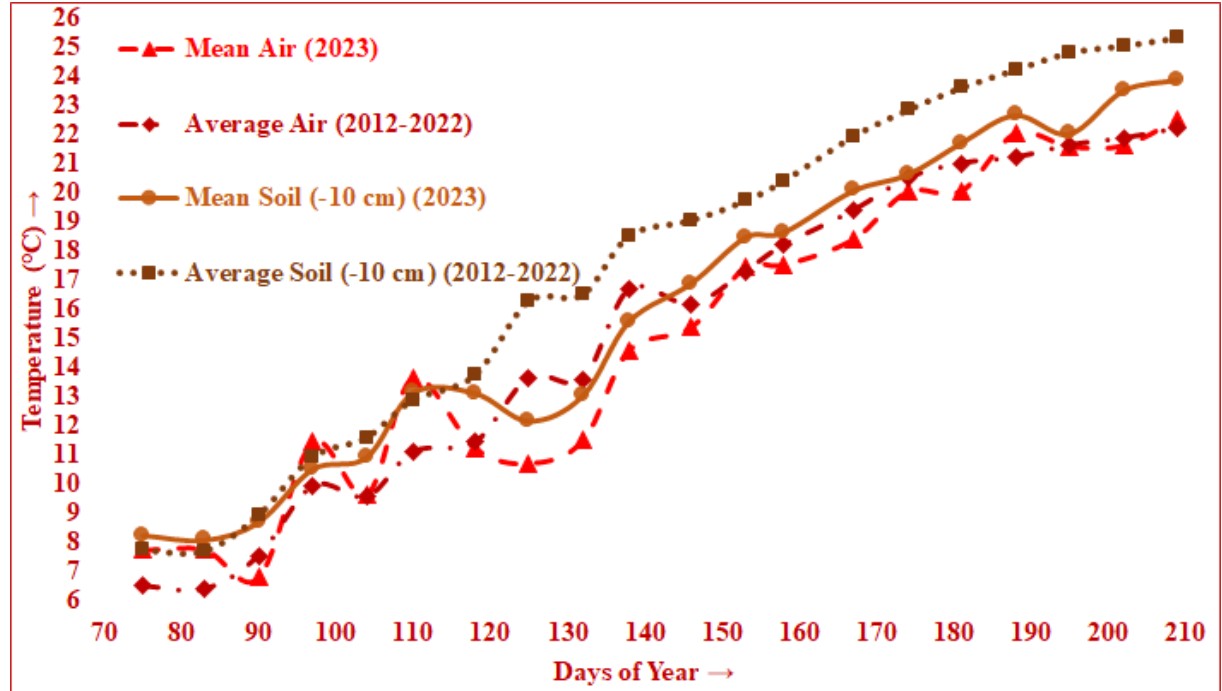

**Figure 4: Mean air and soil (-10 cm) temperature data (2023) together with average air and soil (-10 cm) temperature data (2012-2022) along the relevant monitoring DOYs.**

## 3 Results

At the beginning of the monitoring (DOY: 75), when the oriental planes were almost leafless, the mean LAI was 0.35 $m^2$ $m^{-2}$, that ranged between 0.28 $m^2$ $m^{-2}$ (1. Point) and 0.42 $m^2$ $m^{-2}$ (7. Point) (Fig. 3). The mean LAI remained almost stable until the end of March (Day: 90), increasing only 0.03 $m^2$ $m^{-2}$, ranging between 0.31 $m^2$ $m^{-2}$ (1. Point) and 0.43 $m^2$ $m^{-2}$ (7. Point) (Fig. 3). During the two weeks between 11th and 24th of March (DOY: 70 to 83), the mean air and soil temperatures (-10 cm) were 7.8°C and 8.2°C respectively, yet the mean air temperature dropped to 6.8°C whereas the mean soil temperature (-10 cm) increased to 8.7°C until the end of March (DOY: 90) (Fig. 4). However, the mean air temperature ascended to 11.5°C and again diminished to 9.7°C while the mean soil temperature (-10 cm) ascended first to 10.5°C and then to 11.0°C respectively during the following two weeks (DOY: 90 to 104) (Fig. 4), when the mean LAI increased first to only 0.41 $m^2$ $m^{-2}$ ranging between 0.33 $m^2$ $m^{-2}$ (1. Point) and 0.46 $m^2$ $m^{-2}$ (9. Point), and then to only 0.44 $m^2$ $m^{-2}$ ranging between 0.36 $m^2$ $m^{-2}$ (1. Point) and 0.49 $m^2$ $m^{-2}$ (5. Point) (Fig. 3).

During only the six days at the beginning of the second half of April (DOY: 104 to 110), the mean air and soil temperatures climbed from 9.7°C to 13.7°C, and from 11.0°C to 13.2°C respectively (Fig. 4), which principally triggered the leaf budburst and the directly associated mean LAI climb from 0.44 $m^2$ $m^{-2}$ (ranging between 0.36 $m^2$ $m^{-2}$ for 1. Point and 0.49 $m^2$ $m^{-2}$ for 5. Point) to 0.80 $m^2$ $m^{-2}$ (ranging between 0.72 $m^2$ $m^{-2}$ for 3. Point and 0.92 $m^2$ $m^{-2}$ for 1. Point) (Fig. 3). Although during the



following three weeks until the 12th of May (DOY: 110 to 132), the mean air temperatures dropped from 13.7°C to 11.3°C and then to 10.7°C, and then fluctuated slightly at 11.5°C, it did not do much to affect the soil (-10 cm) temperatures (~12.8°C) (Fig. 4) and the increasing trend of the mean LAI from 0.80 m² m⁻² (0.72 m² m⁻² for 3. Point-0.92 m² m⁻² for 1. Point) to 1.36

190 m² m⁻² (1.07 m² m⁻² for 3. Point-1.51 m² m⁻² for 1. Point) to 1.83 m² m⁻² (1.58 m² m⁻² for 3. Point-2.28 m² m⁻² for 4. Point) to 2.15 m² m⁻² (1.81 m² m⁻² for 3. Point-2.50 m² m⁻² for 6. Point), which indicated the period of the occurrence, flushing and, to some extent expansion of the oriental plane leaves (Fig. 3). The increment in number and size of the oriental plane leaves proceeded during the following three weeks until early-June (DOY: 132 to 153), during when the mean air and soil (-10 cm) temperatures increased from 11.5°C to 17.5°C, and from 13.1°C to 18.5°C to respectively (Fig. 4). All these increments led

the mean LAI rise from 2.15 m² m⁻² to 2.72 m² m⁻² (2.35 m² m⁻² for 3. Point-3.10 m² m⁻² for 4. Point) in 2nd of June (DOY: 153), when the stationary period for the oriental plane leaves almost started (Fig. 3).

During the stationary period principally beginning in early June (DOY: 153) and approximated to have lasted at least until late-July (DOY: 209), the maximum value; 2.76 m² m⁻² of the mean LAI (ranging between 2.41 m² m⁻² for 3. Point and 3.16 m² m⁻² for 4. Point) was achieved (DOY: 158-167), and then it gradually diminished to 2.54 m² m⁻² (ranging between 2.20 m²

m⁻² for 3. Point and 2.84 m² m⁻² for 4. Point) (Fig. 3). However, both the mean air and soil (-10 cm) temperatures gradually continued to increase from 17.6°C to 22.5°C, and from 18.7°C to 23.9°C respectively (DOY: 158 to 209) (Fig. 4). Thus, even though both the mean air and soil (-10 cm) temperatures had apparent influences on the periodical course of the LAIs until their maximum values, their instant decrease for a certain period afterwards, could not immediately and solely be attributed to course of the temperatures belonging to that certain period.

Comparing the maximum values of the LAI (m² m⁻²) for each point, the order of the points was 4. (3.16), 6. (3.02), 1. (2.94), 2. (2.93), 10. (2.89), 7. (2.64), 5. (2.63), 8. (2.60), and 3-9. (2.42) ranking from the highest to the lowest. On the other hand, that order was 6. (26.7; 23.5-31.0), 5. (24.7; 17.5-28.5), 7. (24.3; 19.5-28.0), 3. (24.2; 19.5-27.5), 4. (23.5; 17.5-29.0), 1. (22.5; 19.5-26.0), 8. (21.4; 18.5-24.5), 2. (20.5; 13.0-26.0), 9. (18.7; 16.0-21.0), 10. (17.0; 14.5-20.3) for the mean height (m; with minimums and maximums) of the trees (Fig 5). In addition, for the mean DBH (cm; with minimums and maximums) of the

trees, the order was 3. (38.2; 18.5-57.0), 6. (36.9; 18.0-64.1), 9. (36.3; 23.0-48.7), 1. (36.0; 26.6-49.3), 8. (33.7; 12.0-52.4), 4. (29.9; 12.8-36.0), 10. (29.9; 15.7-46.7), 5. (29.7; 10.7-47.2), 2. (29.5; 14.8-49.8), 7. (26.5; 7.3-40.3) (Fig 5). In fact, the point-based ranking compatibility of the maximum LAI values was valid neither with the mean (minimum-maximum also) height nor with the mean (minimum-maximum also) DBH of the trees (Fig. 5). This situation indicated the maximum LAI values were not directly related either with the height or with the DBH of the trees within the canopy frame of each point. Furthermore,

that ranking order was 1. (15), 3-7. (14), 2-5-6. (12), 4. (11), 8-10. (8), 9. (5) for the number of trees within the canopy frames of the relevant points (Fig. 5). Therefore, it can be deduced that the maximum LAI values did not coincide with these number of the trees, revealing their unremarkable influence on those maximum LAI values (Fig. 5).



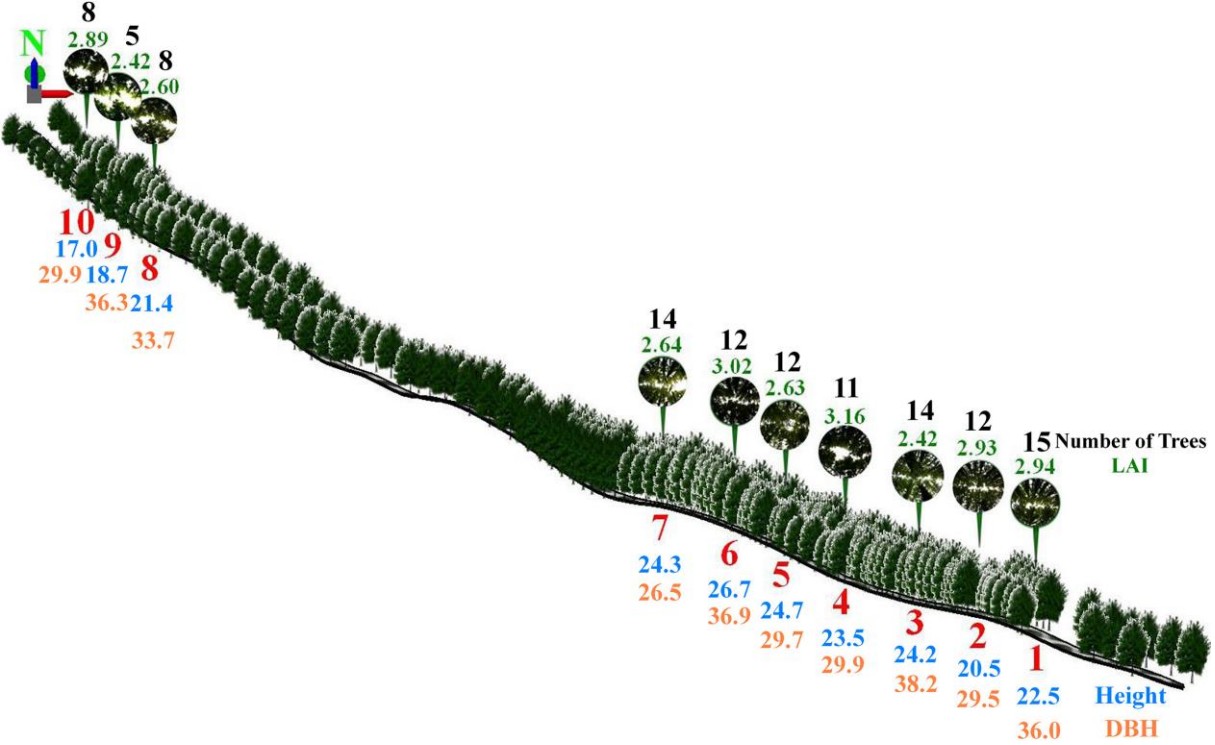

**Figure 5: Exact scaled 3D simulation and illustration of the study field oriental plane trees on the 10 points together with their dense-canopy hemispherical photographs. Also, the number of trees within those canopy frames of each 10 points, the average height (m) and DBH (cm) of those trees, and the maximum LAI values of each of those 10 canopies.**

## 4 Discussion

After the initial monitoring (DOY: 75) of the LAI, the leafless status of the oriental planes remained approximately one month period, which consisted of totally five hemispherical photographing visits to the study field. During this one-month period, the increment of the mean LAI could not reach even 0.1 $m^2$ $m^{-2}$, when their leaf buds had swollen but not burst at least until the mid-April (DOY: 104) (Fig. 3). In their relatively recent study, Sadeghi et al. (2018) monitored the same species; *Platanus orientalis* L. trees, along the whole three years, during when their leafless period had ended approximately in the late-March (second half of March), which was about one-month earlier than our study. Although in that study, altitude of the site was 1000 m higher than our site, basically the latitudinal and longitudinal differences and associated temperature differences with our study (average annual temperature has been 4.6°C higher than our), have led to such one-month phenological earliness for the same tree species (*Platanus orientalis* L.). However, during the 35 days before the mid-April (DOY: 70 to 104), the mean air temperature was 8.7°C, which widely ranged between 0.4°C and 15.4°C (Fig. 4).

On the other hand, essential transformation actually occurred with the budburst of the oriental plane leaves during the following six days in mid-April (DOY: 104 to 110), which resulted in the latest mean LAI, almost doubled (Fig. 3). Conformably, in



their study within a Western-European deciduous forest composed of mature sessile oaks and understorey European hornbeams, Soudani et al. (2021) indicated approximately the same dates as ours for the budburst of their leaves, which also led their mean LAI likely doubled. During these six days, the mean air and soil temperatures (-10 cm) suddenly increased by 4.0°C and by 2.4°C respectively (Fig. 4), which then; until the mid-May (DOY: 110 to 138), led to the onset of the flushing of fresh leaves, and associated increment of the mean LAI approximately 3 times (from 0.80 $m^2$ $m^{-2}$ to 2.39 $m^2$ $m^{-2}$) (Fig. 3). For a temperate deciduous forest, mainly composed of sugar maples and North American beeches, in Ontario, Canada, Croft et al. (2014) determined almost mid-May for the initiation of the LAI increment following the budburst, and almost the very beginning of June for the maximum LAI values. Even though sharing almost the exact average annual precipitation (1050 mm) with our study, average annual air temperature difference of 7.5°C (their 5°C, our 12.5°C) could have led to that one-month-delay for the initiation of LAI increment, compared to our study. However, during that almost one-month period (DOY: 110 to 138), the mean air and soil (-10cm) temperatures continued to climb by 1.0°C and by 2.5°C respectively (Fig. 4). On the other hand, Zolles et al. (2021) detected almost the same dates (DOY: 108) with our study for the unfolding of the European beech trees within stand of deciduous forests at close altitudes in Central Europe.

Furthermore, during the following almost three weeks until early-June (DOY: 138 to 158), the mean air temperature ascended by 2.9°C whereas the soil temperature ascended by 3.1°C (Fig. 4). This situation emphasized the essential role of the soil temperature both on the phenological processes and on the canopy LAI of the oriental plane trees. Thus, in their modelling study, Klinek et al. (2023) suggested that a model based on soil-air temperature could be used to predict the initiation of seasonal phenology of deciduous forests. As a matter fact, in our study, high and significant positive correlations between the mean soil temperatures and the mean LAI values were valid throughout both the leaf budburst-development period (DOY: 104 to 158; *r=0.84, P<0.01*) as well as being valid throughout the entire monitoring periods (DOY: 70 to 209; *r=0.89, P<0.01*) (see also Figs 3 and 4). Although not as much as soil temperature, high and significant positive correlations were also valid between the mean air temperatures and the mean LAI values throughout both the leaf budburst-development period (DOY: 104 to 158; *r=0.75, P<0.03*) and the entire monitoring periods (DOY: 70 to 209; *r=0.85, P<0.01*) (see also Figs 3 and 4). Similarly, within a relatively close study area, Öztürk et al. (2015) revealed the influence and significance of the soil temperature rather than the air temperature on the LAI during the leaf budburst-development period for an urban forest patch composed of European hornbeams. The development of leaves in size and number continued along these following three weeks (DOY: 138 to 158) when the mean LAI increased by 0.37 $m^2$ $m^{-2}$ reaching 2.76 $m^2$ $m^{-2}$; the maximum, and remained constant at this value at least 10 days more (until mid-June; DOY: 167) (Fig. 3).

Point-based correlation analysis between the maximum LAI values and the mean height of the trees, which are included within the canopy frame of the relevant points, showed inexistence of pronounced correlation (DOY: 158 and 167; *r=0.12, P=0.739*) among them (see also Fig. 5). However, for an urban forest in India, Behera et al. (2022) demonstrated high and significant correlation (r=0.95, *P<0.01*) between the LAI and tree height. Besides, in our study, the correlation between the maximum LAI values and the mean DBH of those trees was not valid (DOY: 158 and 167; *r=-0.27, P=0.457*) also (see also Fig. 5). Indeed, Cai et al. (2022) monitored the canopy structures of the roadside trees including also the same oriental planes in



summer, and determined relatively weak and insignificant correlation (*r=-0.20, P>0.05*) between the summer LAI values and the DBH of the urban trees. On the other hand, in their relatively very recent study for the *Ginkgo biloba* and *Platanus orientalis* trees in parks and roadsides of Beijing, China, Cui et al. (2022) correlated both the LA (leaf area) and SLA (Specific Leaf Area) with both the H (height) and DBH (Diameter at Breast Height). They found significant positive correlations between the LA and both the H (*r=0.83, 0.54, P<0.01*) and DBH (*r=0.63, 0.65, P<0.001*) for both of the oriental planes and gingkos

respectively, whereas the correlations between the SLA and both the H (*r=0.27, 0.20, P>0.05*) and DBH (*r=0.35, 0.23, P>0.05*) were relatively weak and insignificant for both of the oriental planes and gingkos.

In our study, the lack of significant correlation between the point-based maximum LAI values and the primary physiological parameters of trees; height and DBH, possible cause of these maximum LAI values could be associated with the number of trees within the canopy frame of each point. Indeed, according to the study by Klingberg et al. (2017), mean effective LAI

based on remote sensing data, was correlated with both the canopy cover and the total tree volume. However, in our study, the number of trees that are included within that canopy frame of the relevant points, was not directly correlated (*r=0.24, P=0.504*) with their maximum LAI values (see also Fig. 5). Beyond and rather than the individual effects of those physiological and quantitative factors, their combined effects together with some possible other ecological and environmental factors have affected the point-based maximum LAI values of these tree canopies. Nevertheless, the difference between the two extreme

values of the point-based maximum LAI, was relatively narrow (± 0.74) (see also Fig. 5).

After achieving the maximum values (DOY: 167 to 209), the gradual yet slightly decrease in the LAI in terms of both the point-based and mean values (Fig. 3), was probably due to some other factors rather than the air-soil temperature. Hence, the mean air-soil temperature values continued to increase gradually until the end of July, 2023 (Fig. 4). These probable factors may involve shortage of soil moisture due to lack of adequate precipitation, windblown leaves. However, the same situation

was valid for another study on the oriental plane trees at another section of the same greenway system about 11 years ago (2012), which also indicated that gradual decrease of the mean and point-based LAI values after the late-June although the mean air temperature similarly continued to increase until the end of July then (Öztürk 2016).

**5 Conclusions**

Although minor differences amongst the LAI values have occurred on the point-basis for each of the definite field visit days,

their mean LAI values showed significant high correlations with the mean temperature data particularly with the soil temperatures throughout the monitoring period of approximately four and half months. This effect of the soil temperature on the oriental plane phenology and on the overall, seasonal and periodical course of the mean LAI values, reveals the significance of the possible warming, that may occur due to the several reasons such as prospective climate warming phenomena, resurfacing practices, and associated newly road asphalt and roadside pavements, heat from wheels and exhaust gases of the

increasing vehicles. These possible situations may therefore indicate the potential alarm of the early budburst dates and associated possible advance of the tree foliation period, which then may lead to the physiological growth of the trees due to




their extended period of photosynthesis. However, in this study, high and significant correlation could not be valid between the height and DBH, and maximum LAI values of those oriental planes along the greenway system.

The consequences of this study reveal the importance of monitoring the foliation course of tree LAI parameter, which can be
significant indicator particularly for the canopy closure, shading, safety and associated recreation on the road and roadside along the outstanding greenway system. Besides, proving the essential role of the air-soil temperature on that LAI course, in reverse suggests the significance of the road and roadside for those alley trees and their canopies. Therefore, this study emphasizes how important sustainability is for the region's most valuable greenway system and somehow implies what the necessary conditions for this sustainability and protection might be. In fact, this relatively old but splendid urban-rural
greenway system with spectacular surrounding forest landscape, is under the possible threat of remaining in the past following the alternation or completely suppression risk after the introduction of the new highway construction campaign within the region.

**Acknowledgments**

Part of the data of this study is from the report of the project "Monitoring Some Canopy Parameters of Trees Throughout Their
Phenological Periods Along a Greenway System Within the Forest Landscape: Greenway System Between The Bartın and Karabük Provinces" submitted to the TUBİTAK (The Scientific and Technological Research Council of Türkiye) and accepted with the number of 2209-A-1919B012223881. Thus, the authors of this study owe TUBİTAK a dept of gratitude for all those financial and encouragement supports. Besides, Forest Engineers Hasan Güneş, Serkan Bulut and Metehan Ayvaz are gratefully acknowledged for their complimentary support during the field works of this study. In addition, we would like to
credit Turkish State Meteorological Service, which provided the meteorological data, and Turkish General Directorate of Forestry, which shared the forest management plans and maps with us. Above all, we would like to appreciate our Bartın University, which continuously encouraged our studies with financial and moral supports.



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





**Statements & Declarations**

**Funding**

Part of this study was supported by the TUBİTAK (The Scientific and Technological Research Council of Türkiye) with the
grant number of 2209-A-1919B012223881. Thus, one of the authors; Rıdvan KORUYAN has received that research support
from the TUBİTAK.

**Competing Interests**

There is no competing interest with any of the parties, which will further risk this study into any controversies. Therefore, the
authors have no relevant financial or non-financial interests to disclose.

**Author Contributions**

All the authors contributed to the study conception and design. On the other hand, all the field works and monitoring processes
were conducted by Melih Öztürk, Turgay Biricik and Rıdvan Koruyan. The parameter and statistical analyses were performed
by Melih Öztürk, Turgay Biricik and Rıdvan Koruyan. The data tabulating, graphics and 3D modelling were done by Melih
Öztürk. The ultimate writing processes were completed by Melih Öztürk, Turgay Biricik and Rıdvan Koruyan.