# Peer review of "Role of air-soil temperature on the LAI course and role of height-DBH on the maximum LAI during foliation of *Platanus orientalis* L. along an urban-rural greenway system"

_EGUsphere, 2024_

## Author Response (AR1)

CommentR1:

Overall Evaluation: The study intends to determine the effect of air and soil temperature on the progress of LAI parameter for Platanus tree. It also examines the effects of some physical parameters of Platanus on its' max LAI. This study is applied within the urban to rural greenway system. The study generally meets its' overall target to supply the importance of the LAI for the scientific audiences, which deal with greenway landscapes and trees. On the other hand because the study includes biological and pedological and to some extent environmental aspects, it fits the concepts and standards of the Biogeosciences Journal. However it requires some revisions to achieve more improvement to be published. They are such as follows:

Answer:

Thanks for these positive evaluations on our study. And necessary revisions were done as follows:

CommentR1: Abstract

Lines 9-12: Please rewrite the sentence that describes aim and context of the study and LAI, since it is already not obvious. To clarify, you may divide the sentence.

Answer:

To clarify this long sentence: "Therefore, particularly during their foliation periods, monitoring and analyses of that phenological and eco-physiological course of these roadside trees referring to the significant and comprehensive canopy parameter; LAI, primarily will indicate their gradual degree of closure, and will determine their gradual coverage on the road and the roadside."

It was divided and rewritten as: "Therefore, particularly during their foliation periods, monitoring and analyses of that phenological and eco-physiological course of these roadside trees primarily will indicate their gradual degree of closure, and will determine their gradual coverage on the road and the roadside. Hence, LAI is a significant and comprehensive canopy parameter which is referred for those monitoring and analyses procedures."

CommentR1: Abstract

Line 17: well-known greenway system? Must be written as regionally well-known greenway.

Answer:

It was written as suggested by the reviewer: "Therefore, for this study, in order to monitor and determine the development and change of the LAI, hemispherical photographs were taken beneath the tree canopies at 10 different points along the part of a regionally well-known greenway system, which involves alleys of Platanus orientalis L. (oriental plane) trees."

CommentR1: Abstract

Lines 23-25: Do not repeat units in parentheses, m2, cm, m... at least for the abstract.

Answer:

Repeated units were deleted correcting the sentences as: "The seasonal course of the LAI values was graphed for each point and principally the daily-based mean LAI (ranging between 0.35 and 2.76 $m^2 m^{-2}$) was evaluated referring to both the air and soil (-10 cm) temperature data. The point-based maximum LAI values (average 2.76 $m^2 m^{-2}$; ranging between 2.42 and 3.16), which were achieved during the mid-June, were examined comparing their ranking order with those of the basic physiological parameters; mean height (ranging between 17.0 and 26.7 m) and mean DBH (ranging between 26.5 and 38.2 cm), and number of trees (5 to 15) within the canopy frames of the relevant points.

CommentR1: Abstract

Keywords: air-soil temp. also must have semicolon.

Answer:

Semicolon were added.

CommentR1: Introduction

Line 39: wooded ecosyt? Do you mean woodland ecosystems?

Answer:

The sentence was corrected according to the suggestion as: "These intercity roads that pass throughout the woodlands and forests mostly constitute rural greenway systems, ultimately become part of the overall rural landscape by integrating with those surrounding woodland ecosystems (Konijnendijk, 2008; Turner and Gardner, 2015)."

CommentR1: Introduction

Line 45: differentiate or determine?

Answer:

Here the word "Differentiate" is more appropriate compared to the word "determine". So we did not make any change.

CommentR1: Introduction

Line 48: "as well" must be at the end of the sentence.

Answer:

According to this suggestion, the sentence was changed as: "Therefore, continuous monitoring on the purpose of determining both their phenological course and physiological characteristics through the analyses of certain vegetation parameters, will help to define the degree of their canopy closure and relevant coverage (Granero-Belinchon, 2020; Pu, 2021) on the road and roadsides of those rural greenway systems as well (Wang et al., 2019)."

CommentR1: Introduction

Line 53: Is "they form" necessary?

Answer:

We deleted the phrase "they form" and new sentence is: "Dependent principally upon their active role on the photosynthesis, leaves and canopies are effective on stem physiology of the trees (Pretzsch, 2009)."

CommentR1: Introduction

Line 58: First reference 'Blanchard-2016' is an example. What about second reference 'Cote', it is also an example, and e.g. must be added?

Answer:

e.g., was also added to the second reference and the sentence became: "Although there are scientific studies that have tried to correlate crown area (e.g., Blanchard et al., 2016) and canopy structure (e.g., Côté et al., 2012) of the trees with their height and DBH, tree canopy parameters that are directly referred as indicators of the canopy characteristics under the influence of those stem physiological characteristics, are relatively restricted."

CommentR1: Introduction

Line 64: Instead of "accordingly", "consequently" may better fit.

Answer:

According to the suggestion the sentence was changed as: "Nonetheless, the LAI parameter is consequently an indicator of canopy closure, and associated shading potential of those tree canopies, ultimately the recreation potential of that environment where those trees exist (Zhang et al., 2022)."

CommentR1: Introduction

Line 72: not "any role", only "role" suits.

Answer:

According to the suggestion, we changed the sentence as: "Consequently, the main aim of this study is to investigate and reveal the priority and effective role of the air-soil temperature on the seasonal change of the LAI trees, and to question the existence of the role of the tree physiology on the maximum LAI values along a roadside."

CommentR1: Introduction

Line 80: not "would have been examined". It will be determined or defined. Therefore "would have been defined" may suits.

Answer:

According to the suggestion, we changed the sentence as: "By this way, the influence of the tree stem physiology on the maximum LAI values of the tree canopies would have been determined."

CommentR1: Introduction

Line 84: "Their" must be "Its'"

Line 85: again "their"

Line 86: "They" or otherwise use plural oriental planes at the beginning of the plant description.

Answers:

According to the suggestion, we changed the following sentences as: "Its' massive trunk is smooth with ash-grey bark, which peels off in large thin flakes whereas its' large leaves with flat surface are cut into five deep lobes, with numerous secondary notches (Johns, 2014). Oriental planes are both forest and ornamental trees, which naturally extend from the Balkans towards the Anatolia, and then towards the Himalayas (Davis, 1982)."

CommentR1: Figure 1: Resolution of the maps should be increased during the publishing.

Answer:

Their original forms with high resolutions will be uploaded ultimately.

CommentR1: Material and Methodology

Study Site Characteristics and Hemispherical Photographing Points

Line 100: The well-known state of greenway system comes from here for abstract. Also long sentence must be divided.

Answer:

According to the suggestion, we changed and divided the following sentence as: "On the other side, the study site is approximately 40 km away from the World Heritage City, Safranbolu at the south (UNESCO, 1994). It is about 50 km away from its' Karabük Province, also at the south, where there are also frequent district and village settlements along the road (Fig. 1)."

CommentR1: Material and Methodology

Line 103: "quite dense" "that are quite dense" should be changed as this.

Answer:

According to the suggestion, we changed the following sentence as: "Therefore, the greenway system along this intercity road, which is famous primarily from the point of spectacular tree canopy tunnels, that are quite dense in some places."

CommentR1: Material and Methodology

Line 115: "foot" must be plural due to the plural form of "hills"

Answers:

"Foot of the hills" is an expression which cannot be written as "feet of the hills". Therefore, we did not make any change.

CommentR1: Material and Methodology

Line 118: To clarify "the lowlands" should be attributed to the "hills" by changing "the lowlands" to "their lowlands". Right?

Answers:

According to the suggestion, we changed the following sentence as: "On the other hand, some agricultural fields, which are principally at the foot of the hills and at their lowlands close to the stream, include few village houses. The altitude of those surrounding hills range between 390 m asl. and 490 m asl. The shallow (20-50 cm) or moderate deep (50-90 cm) grey-brown podzolic soils together with brown forest soils have formed particularly on the slope and top of the hills whereas deep (90+ cm) alluvial soils have been deposited at their lowlands close to the stream (TMAF, 2005)."

CommentR1: Figure 2: Resolution may be increased more.

Answer:

Their original forms with high resolutions will be uploaded ultimately.

CommentR1: Meteorological Calculations

Line 131: Specify the scientific literature.

Line 133: Refer Barry-2008 for the previous scientific literature specification previously.

Answers:

According to the suggestions, we changed the following sentence as: "The data of the six different meteorological stations, which were at the 36 m asl. (~33 km away at northwest; Bartın), at the 134 m asl. (~12 km away at northwest; Ulus), at the 359 m asl. (~5 km away at east; Ovacuma), at 520 m asl. (~12 km away at west; Ceyüpler), at the 540 m asl. (~29 km away at northeast; Çubukeli), at the 1090 m asl. (11 km away at south; Orman), were referred during the analyses and determination of the air temperature values for the study field as suggested by the scientific literature (Barry, 2008) for such as this mesothermal humid region (TSMS, 2022)."

CommentR1: Meteorological Calculations

Line 131: "initial correlation analyses" more correct?

Answer:

According to the suggestion, we changed the following sentence as: "As a matter of fact, initial correlation analyses were conducted between the air temperature data of these meteorological stations."

CommentR1: Meteorological Calculations

Line 133: Is "correlation" necessary in the parenthesis?

Answer:

According to the suggestion, we changed the following sentence as: "Some of their correlation (r=0.96; P<0.01) and regression suggested the lapse rate of 0.5°C/100 m asl., as also suggested by Barry (2008) for the mesothermal humid regions."

CommentR1: Meteorological Calculations

Line 134: Soil temperature data measured in situ? Or referred from the meteorological station. It is unclear according to the sentences. Please clarify.

Answer:

According to the suggestion, to clarify it, we changed the following sentence as: "Since its' air temperature data were more consistent and soil temperature data were also measured at that station, the temperature data of Bartın Metetorological Station were used during the calculation of the temperature data for the study field (TSMS, 2022)."

CommentR1: Meteorological Calculations

Line 138: Add "soil depth" after the 15 cm. Or add minus before numbers.

Answer:

According to the suggestion, we changed the following sentence as: "Dependent upon the densely distribution of the oriental plane tree roots within the 0 and 15 cm soil depth (Xie et al., 2020), the soil temperature data belonging to the -10 cm soil depth were referred."

CommentR1: Meteorological Calculations

Line 141: Move "were calculated" after the "tempt. data". Otherwise sentence is not clear.

Answer:

According to the suggestion, we changed the following sentence as: "Provided that the 1-week air-soil (-10 cm) temperature data prior to the first monitoring in early-March were included, the modified mean air-soil temperature data were calculated throughout those frequent intervals between each field visits for hemispherical photographing."

CommentR1: Hemispherical Photographing and LAI, Height, DBH Analyses

Line 146: Is writing "middle-lane" proper?

Line 148: "middle-lane"

Answer:

According to the suggestion, we changed the following sentence as: "The 10 hemispherical photographing points on the middle-lane of the road were chosen in the field and marked (Fig. 1). The hemispherical photographing (180°) of the tree canopies were conducted on these fixed points beneath and towards those canopies above. For this hemispherical photographing perpendicular (90°) to the middle-lane of the road, an 8 mm fisheye objective (Sigma F3.5 EX DG Circular Fisheye-Sigma Corporation) was mounted on a digital camera (Canon EOS 5D SLR digital camera-Canon Inc.)"

CommentR1: Hemispherical Photographing and LAI, Height, DBH Analyses

Line 151: "were conducted"?

Answer:

In order to avoid the more repetition of the word "conducted", we kept the phrase "were done".

CommentR1: Hemispherical Photographing and LAI, Height, DBH Analyses

Line 151: "using" instead of "by"?

Answer:

According to the suggestion, we changed the following sentence as: "Hence, in order to analyse the LAI values, the digital images of these 200 hemispherical photographs were processed using the software program; Hemisfer version 3.3 (Swiss Federal Institute of Forest, Snow and Landscape Research; Schleppi et al., 2007)."

CommentR1: Hemispherical Photographing and LAI, Height, DBH Analyses

Line 158: "when they were foliated" refers to specific time or refers to fully-foliated period? Please clarify the sentence.

Answer:

According to the suggestion, we changed the following sentence as: "Thereby, the height and DBH of each of these plane trees within those frames, were measured and recorded according to their points, when they were fully-foliated."

CommentR1: Hemispherical Photographing and LAI, Height, DBH Analyses

Lines 161-165: Separate the statistical analyses to another paragraph.

Answer:

According to the suggestion, we moved the statistical analyse to the next paragraph.

CommentR1: Figures 3 and 4: Image resolutions may be improved.

Answer:

Their original forms with high resolutions will be uploaded ultimately.

CommentR1: Results

Lines 183-186: The sentence that starts with "During only" is long and should be divided to be more explicit.

Answer:

According to the suggestion, we changed and divided the following sentences as: "During only the six days at the beginning of the second half of April (DOY: 104 to 110), the mean air and soil temperatures climbed from 9.7°C to 13.7°C, and from 11.0°C to 13.2°C respectively (Fig. 4). These temperatures principally triggered the leaf budburst and the directly associated mean LAI climb from 0.44 m2 m-2 (ranging between 0.36 m2 m-2 for 1. Point and 0.49 m2 m-2 for 5. Point) to 0.80 m2 m-2 (ranging between 0.72 m2 m-2 for 3. Point and 0.92 m2 m-2 for 1. Point) (Fig. 3)."

CommentR1: Results

Lines 187-192: The following sentence is also long that includes much relative clauses. The sentence may be broken into two from these relative clauses.

Answer:

According to the suggestion, we changed and divided the following sentences as: "Although during the following three weeks until the 12th of May (DOY: 110 to 132), the mean air temperatures dropped from 13.7°C to 11.3°C and then to 10.7°C, and then fluctuated slightly at 11.5°C, it did not do much to affect the soil (-10 cm)  temperatures (~12.8°C) (Fig. 4) and the increasing trend of the mean LAI from 0.80 m2 m-2 (0.72 m2 m-2 for 3. Point-0.92 m2 m-2 for 1. Point) to 1.36 m2 m-2 (1.07 m2 m-2 for 3. Point-1.51 m2 m-2 for 1. Point) to 1.83 m2 m-2 (1.58 m2 m-2 for 3. Point-2.28 m2 m-2 for 4. Point) to 2.15 m2 m-2 (1.81 m2 m-2 for 3. Point-2.50 m2 m-2 for 6. Point). This situation indicated the period of the occurrence, flushing and, to some extent expansion of the oriental plane leaves (Fig. 3)."

CommentR1: Figure 5: Image resolution improvement is necessary.

Answer:

Their original forms with high resolutions will be uploaded ultimately.

CommentR1: Discussion

Line 228: "along the whole three years". Avoid the word "whole".

Answer:

According to the suggestion, we changed the following sentences as: "In their relatively recent study, Sadeghi et al. (2018) monitored the same species; Platanus orientalis L. trees, along the three years, during when their leafless period had ended approximately in the late-March (second half of March), which was about one-month earlier than our study."

CommentR1: Discussion

Line 238: "likely doubled" should be replaced by "similarly doubled.

Answer:

Here the word "likely" does not refer to the similarity but refers to the approximation. Therefore, it remained as is.

CommentR1: Discussion

Line 238-240: The sentence may be divided from the "which then;". Continue the next sentence with only "Then".

Answer:

According to the suggestion, we changed and divided the following sentences as: "During these six days, the mean air and soil temperatures (-10 cm) suddenly increased by 4.0°C and by 2.4°C respectively (Fig. 4). Then, until the mid-May (DOY: 110 to 138), this led to the onset of the flushing of fresh leaves, and associated increment of the mean LAI approximately 3 times (from 0.80 m2 m-2 to 2.39 m2 m-2) (Fig. 3)."

CommentR1: Discussion

Line 243: "almost the very beginning of June" may just be "almost early-June" Right?

Answer:

According to the suggestion, we changed the following sentences as: "For a temperate deciduous forest, mainly composed of sugar maples and North American beeches, in Ontario, Canada, Croft et al. (2014) determined almost mid-May for the initiation of the LAI increment following the budburst, and almost early-June for the maximum LAI values."

CommentR1: Discussion

Line 248: Is the DOY: 108 for the unfolding or the onset of the unfolding? Because leaf unfolding is a relatively lasting period rather than an exact date. Also "unfolding" necessitates to be "leaf unfolding" to be more specific.

Answer:

According to the suggestion, we changed the following sentences as: "On the other hand, Zolles et al. (2021) detected almost the same dates (around DOY: 108) with our study for the leaf unfolding of the European beech trees within stand of deciduous forests at close altitudes in Central Europe."

CommentR1: Discussion

Line 268: Take the "also" from the end of the sentence after "trees was".

Answer:

According to the suggestion, we changed the following sentences as: "Besides, in our study, the correlation between the maximum LAI values and the mean DBH of those trees was also not valid (DOY: 158 and 167; r=-0.27, P=0.457) (see also Fig. 5)."

CommentR1: Discussion

Line 278: The paragraph sentence that starts with "In our study" sounds wrong according to its' structure. Rewrite the sentence.

Answer:

According to the suggestion, we changed the following sentences as: "In our study, due to the lack of significant correlation between the point-based maximum LAI values and the primary physiological parameters of trees; height and DBH, possible cause of these maximum LAI values could be associated with the number of trees within the canopy frame of each point."

CommentR1: Discussion

Line 292: Check the referencing styles for the last sentence and also for the whole text to avoid incompatibility.

Answer:

According to the suggestion, we changed the following sentences as: "However, the same situation was valid for another study on the oriental plane trees at another section of the same greenway system about 11 years ago (2012), which also indicated that gradual decrease of the mean and point-based LAI values after the late-June although the mean air temperature similarly continued to increase until the end of July then (Öztürk, 2016)."

Also other necessary controls were done.

CommentR1: Conclusions

Line 300: "These possible situations" must be more specific by including the word "threatening" before the situations phrase.

Answer:

According to the suggestion, we changed the following sentences as: "These threatening possible situations may therefore indicate the potential alarm of the early budburst dates and associated possible advance of the tree foliation period, which then may lead to the physiological growth of the trees due to their extended period of photosynthesis."

CommentR1: Conclusions

Line 303: "could not be valid" suggests somehow anticipation. "was not valid" may be enough.

Answer:

According to the suggestion, we changed the following sentences as: "However, in this study, high and significant correlation was not valid between the height and DBH, and maximum LAI values of those oriental planes along the greenway system."

CommentR1: Conclusions

Line 308: Instead of "what the necessary conditions" "how the necessary conditions".

Answer:

According to the suggestion, we changed the following sentences as: "Therefore, this study emphasizes how important sustainability is for the region's most valuable greenway system and somehow implies how the necessary conditions for this sustainability and protection might be."

CommentR1: Conclusions

Line 308: Instead of "what the necessary conditions" "how the necessary conditions".

Answer:

According to the suggestion, we changed the following sentences as: "Therefore, this study emphasizes how important sustainability is for the region's most valuable greenway system and somehow implies how the necessary conditions for this sustainability and protection might be."

ReferencesR1

I will advise to check their spelling both here and within the text once again.

Answer:

All the necessary corrections are made.

CommentR2:

Greenways are important issues for the landscape architecture. They are also valuable for the other ecological sciences. Analysis of these greenways contribute much to the landscape architecture studies. So they indicate also role of the landscape trees for the ecology and the society. Thus this study evaluates urban-rural greenways focusing on their some tree parameters.

Following reviews must be done to make this study more complete for the scientific literature.

Answer:

Thanks for these positive evaluations on our study. And necessary revisions were done as follows:

CommentR2: Abstract

Well-written but some long sentences must be shortened such as line 10 and 23.

Also check the other sentences to shorten if necessary.

Answer:

It (line 10) was divided and rewritten as: "Therefore, particularly during their foliation periods, monitoring and analyses of that phenological and eco-physiological course of these roadside trees primarily will indicate their gradual degree of closure, and will determine their gradual coverage on the road and the roadside. Hence, LAI is a significant and comprehensive canopy parameter which is referred for those monitoring and analyses procedures."

It (line 23) was divided and rewritten as: "The point-based maximum LAI values (average 2.76 m2 m-2; ranging between 2.42 m2 m-2 and 3.16 m2 m-2) were achieved during the mid-June. They were examined comparing their ranking order with those of the basic physiological parameters; mean height (ranging between 17.0 m and 26.7 m) and mean DBH (ranging between 26.5 cm and 38.2 cm), and number of trees (5 to 15) within the canopy frames of the relevant points."

CommentR2: Introduction

Prepared with details but line 48 is also long.

Answer:

The sentence changed as: "Therefore, continuous monitoring on the purpose of determining both their phenological course and physiological characteristics through the analyses of certain vegetation parameters, will help to define the degree of their canopy closure and relevant coverage (Granero-Belinchon, 2020; Pu, 2021) on the road and roadsides of those rural greenway systems as well (Wang et al., 2019)."

Yet, it was not shortened to sustain the continuity of the sentence.

CommentR2: Introduction

Reference studies are not recent all as line 65. Write relatively recent for them.

Answer:

Although there are more recent studies which investigate the relation between the LAI and tree heights and/or canopy heights, studies that have found high and significant positive correlations between them are very scarce. Therefore, the provided references are the most recent ones among those parameters.

CommentR2: Introduction

What is max LAI date in line 75? Write explicit.

Answer:

According to this suggestion, the sentence was changed as: "Thus, in this study, LAI course of the oriental plane (Platanus orientalis L.) trees was monitored and analysed principally throughout the foliation seasons particularly referring to the daily-based relevant temperature data and referring to their physiological parameters during solely the dates when the maximum LAI values were achieved."

CommentR2: Site Characteristics and Points:

UNESCO 1994 is not in the references. References: Add UNESCO 1994 here.

Answer: The following reference was included into the References section as web site of UNESCO:

UNESCO: World Heritage Convention Web Site. City of Safranbolu, https://whc.unesco.org/en/list/614/. 1994

CommentR2: Material and Methodology

The sentence in line 103 is not a complete sentence. There is not a verb. Therefore greenway system....

Answer:

According to the suggestion, we changed the following sentence as: "Therefore, the greenway system along this intercity road, which is famous primarily from the point of spectacular tree canopy tunnels, are quite dense in some places."

CommentR2: Material and Methodology

Monitoring studies for which purposes? Write explicit in line 104.

Answer:

To be explicit, the sentence was changed as "In order to secure the sustainability of this greenway system, road construction and repair works are carried out very sensitively, and hence environmental and ecological monitoring studies are encouraged."

CommentR2: Material and Methodology

You can also refer to the figures for agricultural areas and villages.... Line 116.

Answer:

According to the suggestion, we changed the following sentence as: "On the other hand, some agricultural fields, which are principally at the foot of the hills and at the lowlands close to the stream, include few village houses (Figs. 1 and 2)."

CommentR2: Hemispherical Photographing and LAI, Height, DBH Analyses

In line 159, during foliation or during fully-foliated when describing hemispherical photographing?

Answer:

According to the suggestion, we changed the following sentence as: "Thereby, the height and DBH of each of these plane trees within those frames, were measured and recorded according to their points, when they were fully-foliated."

CommentR2: Results
Long sentences must be shortened. Because results must be clear.

Answer: We changed and divided the following sentences as: "During only the six days at the beginning of the second half of April (DOY: 104 to 110), the mean air and soil temperatures climbed from 9.7°C to 13.7°C, and from 11.0°C to 13.2°C respectively (Fig. 4). These temperatures principally triggered the leaf budburst and the directly associated mean LAI climb from 0.44 m2 m-2 (ranging between 0.36 m2 m-2 for 1. Point and 0.49 m2 m-2 for 5. Point) to 0.80 m2 m-2 (ranging between 0.72 m2 m-2 for 3. Point and 0.92 m2 m-2 for 1. Point) (Fig. 3)."

We changed and divided the following sentences as: "Although during the following three weeks until the 12th of May (DOY: 110 to 132), the mean air temperatures dropped from 13.7°C to 11.3°C and then to 10.7°C, and then fluctuated slightly at 11.5°C, it did not do much to affect the soil (-10 cm) temperatures (~12.8°C) (Fig. 4) and the increasing trend of the mean LAI from 0.80 m2 m-2 (0.72 m2 m-2 for 3. Point-0.92 m2 m-2 for 1. Point) to 1.36 m2 m-2 (1.07 m2 m-2 for 3. Point-1.51 m2 m-2 for 1. Point) to 1.83 m2 m-2 (1.58 m2 m-2 for 3. Point-2.28 m2 m-2 for 4. Point) to 2.15 m2 m-2 (1.81 m2 m-2 for 3. Point-2.50 m2 m-2 for 6. Point). This situation indicated the period of the occurrence, flushing and, to some extent expansion of the oriental plane leaves (Fig. 3)."

CommentR2: Results

In line 197, approximated to have lasted? Is it approximately to have....?

According to the suggestion, we changed the sentence as: "During the stationary period principally beginning in early June (DOY: 153) and approximately to have lasted at least until late-July (DOY: 209), the maximum value; 2.76 m2 m-2 of the mean LAI (ranging between 2.41 m2 m-2 for 3. Point and 3.16 m2 m-2 for 4. Point) was achieved (DOY: 158-167), and then it gradually diminished to 2.54 m2 m-2 (ranging between 2.20 m2 m-2 for 3. Point and 2.84 m2 m-2 for 4. Point) (Fig. 3)."

CommentR2: Discussion

Well-discussed and referenced but correct some sentences when passing between referenced studies and your study.

Answer:

We changed and divided the following sentences as: "During these six days, the mean air and soil temperatures (-10 cm) suddenly increased by 4.0°C and by 2.4°C respectively (Fig. 4). Then, until the mid-May (DOY: 110 to 138), this led to the onset of the flushing of fresh leaves, and associated increment of the mean LAI approximately 3 times (from 0.80 m2 m-2 to 2.39 m2 m-2) (Fig. 3)."